# Scanning nuclear resonance imaging of a hyperfine-coupled quantum Hall system

Katsushi Hashimoto [1,2], Toru Tomimatsu[1], Ken Sato[1] & Yoshiro Hirayama[1,2]

Nuclear resonance (NR) is widely used to detect and characterise nuclear spin polarisation and conduction electron spin polarisation coupled by a hyperfine interaction. While the macroscopic aspects of such hyperfine-coupled systems have been addressed in most relevant studies, the essential role of local variation in both types of spin polarisation has been indicated in 2D semiconductor systems. In this study, we apply a recently developed local and highly sensitive NR based on a scanning probe to a hyperfine-coupled quantum Hall (QH) system in a 2D electron gas subject to a strong magnetic field. We succeed in imaging the NR intensity and Knight shift, uncovering the spatial distribution of both the nuclear and electron spin polarisation. The results reveal the microscopic origin of the nonequilibrium QH phenomena, and highlight the potential use of our technique in microscopic studies on various electron spin systems as well as their correlations with nuclear spins.

[1] Graduate School of Sciences, Tohoku University, Sendai 980-8578, Japan. [2] Centre for Spintronics Research Network, Tohoku University, Sendai 980-8578, Japan. Correspondence and requests for materials should be addressed to K.H. (email: hashi@tohoku.ac.jp)

Quantum Hall (QH) systems in 2D electron gas (2DEG) under a strong magnetic field comprise a wide variety of electronic spin states that often couple with nuclear spins of the host material through hyperfine interactions. Hyperfine-coupled QH systems, such as selectively injected spin-resolved edge channels[1], the QH ferromagnet at the Landau-level (LL) filling factor $v = 2/3$[2–5] and the QH breakdown regime[6], allows highly sensitive resistive detection (RD) of nuclear resonance (NR) as well as current-driven dynamic nuclear polarisation (DNP), which further enhances RD sensitivity. Furthermore, RD-NR spectroscopy, particularly in GaAs-based samples, provides us with an essential probe of electronic spin polarisation in the 2DEG. This technique utilises the hyperfine Fermi-contact interaction between the magnetic moments of nuclei in a GaAs quantum well and the s-type conduction band electrons in the 2DEG. A non-zero electron spin polarisation modifies the effective magnetic field experienced by the nuclei, leading to shifts in the NR frequency. The absolute value of this NR spectrum shift, instead of the ordinal relative value[7], is used as the Knight shift proportional to the electron spin polarisation for direct measurement of the electron spin polarisation in various exotic QH states such as the QH ferromagnet at $v = 1$[8], (and also $2/3$[9]), $v = 5/2$ quantum fluid[10], and the canted antiferromagnet in the bilayer $v = 2$[11].

On one hand, although RD senses macroscopic spin polarisation of the nuclei and electrons, detailed RD-NR investigations have highlighted crucial characteristics of non-uniform distributions, including those of the current-induced DNP in the QH ferromagnet[12] and QH breakdown[6], and of the electron-spin polarisation/density in the QH Wigner crystal[13] and QH stripe phase[14]. However, to fully understand its microscopic origins, there is a strong demand for direct observation of the spin distribution. A recent imaging technique based on scanning photoluminescence microscopy within the particular QH $v = 2/3$

domain system visualised the electron spin distributions of the domains[15], the NR response of which provided spatial information of the DNP[16]. The optical measurement, however, provides the influence of photo-excited carriers and has a limited spatial resolution. More versatile electrical imaging tools based on scanning probes, which are widely used to visualise quantum aspects at higher spatial resolution[17–21], constitute a candidate technology for microscopic NR imaging in semiconductor quantum systems; however, these have not been tested yet.

In this article, we show the spatial distribution of the DNP and electron spin polarisation using a scanning probe-based NR microscope. Our model of a hyperfine-coupled system is the nonequilibrium $v = 1$ QH state driven by imposed electric current. Owing to the accompanying electric field imposed on the 2DEG, the nonequilibrium electrons are excited from the energetically lower spin LL (LL0↑) to the upper (LL0↓) by, for example, inter-LL scattering, which causes QH breakdown. The accompanying electron-spin flip transfers its angular momentum to the nuclei and thereby drives the DNP[6].

## Results

**Local NR.** Figure 1a illustrates our setup[22] based on the methodology of the scanning gate microscope[19]. We used 10-μm-wide Hall bar samples (A, B), which were fabricated from a 20-nm-wide GaAs/Al$_{0.3}$Ga$_{0.7}$As quantum well wafer with a mobility of ~130 m$^2$ V$^{-1}$ s$^{-1}$ at an electron density of $2 \times 10^{15}$ m$^{-2}$. For local NR excitation (see Methods section), a scanning metallic tip was used to apply an RF electric ($E$) field to nuclear spins in the GaAs quantum well located 165 nm below the surface. The RF $E$ field dynamically distorts the electronic environment of the nucleus. This induces the time-varying $E$-field gradient, which can couple[23,24] with the quadrupolar

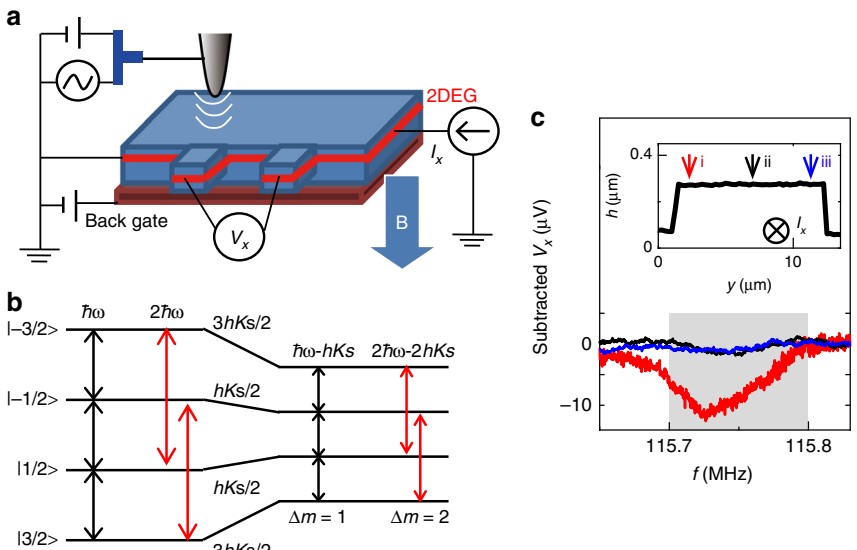

**Fig. 1** Local NR measurements. **a** Schematic experimental setup of the scanning NR microscope. The RF $E$-field-mediated NR is excited by applying RF voltage to a metallic tip with a continuously pumping current ($I_x$)-induced DNP in the QH breakdown regime near $v = 1$, which was tuned by the $B$-field and back-gated electron density. NR is detected using the resistive detection technique by measuring the longitudinal voltage $V_x$. **b** Schematic energy-level diagram of the nuclear spin states of $^{75}$As for $I = 3/2$ with (right) and without (left) the Knight shift ($K_s$). The RF $E$ field at twice the NR frequency induces, instead of $\Delta m = \pm 1$ transitions (black arrows), $\Delta m = \pm 2$ transitions (red arrows)[23,24] through electric quadrupolar coupling. The Knight shift for $2f_{^{75}As}$ also increases to twice the value $2K_s$. **c** Local $^{75}$As NR spectra captured using a sweeping frequency of the continuous-wave $E$ field at a sweeping rate of 330 Hz/s from high to low frequency and RF power of −10 dBm at $B = 8$ T and $I_x = 0.6$ μA in the $v = 1.05$ QH breakdown (marked in the $V_x$-$I_x$ curve shown in Supplementary Note 4). The spectra i, ii, and iii were captured at the points marked on the topographic profile of the Hall bar (inset). The obtained spectra were displayed in the corresponding colours after subtracting the position-dependent $V_x$ offset from the raw spectra

nuclear spins such as the total spin $I = 3/2$ for $^{75}$As, $^{69}$Ga, and $^{71}$Ga nuclei and eventually leads to the NR (Supplementary Note 1). This $E$-field-mediated NR can be driven at not only the fundamental NR frequency, but also at twice this frequency. The latter NR involves $\Delta m = \pm 2$ transitions[23,24] (red arrows) instead of $\Delta m = \pm 1$ transitions (black arrows) in the $I = 3/2$ levels (Fig. 1b), labelled by the magnetic quantum number, $m = \pm 1/2$, $\pm 3/2$. In our local NR measurements, the NR at twice the resonance frequency was exploited due to its advantage of higher spatial resolution[22]. The locally driven NR was, in turn, resistively detected[1–6,8–14] by measuring the longitudinal voltage, $V_x$ at constant $I_x$. This detection scheme relies on the Overhauser field of nuclear spin polarisation, which modifies electronic Zeeman splitting, and hence, the resistance in the spin-split QH regimes (see Methods section). The scanning NR microscope was mounted on a dilution refrigerator with a superconductor magnet. All measurements were performed by pumping the current-induced DNP near $\nu = 1$ QH breakdown at a magnetic field ($B$) of 7–8 T and a sample temperature below 300 mK.

Figure 1c shows the raw data of $^{75}$As NR spectra recorded by the sweeping continuous-wave frequency of the RF $E$ field at $B = 8$ T and $\nu = 1.05$ QH breakdown condition marked in the $V_x$–$I_x$ curve (Supplementary Fig. 2). At the point i near the left-side edge of the Hall bar (inset of Fig. 1c), an NR signal (red curve) was clearly observed at a frequency of ~115.725 MHz with asymmetric broadening [indicated grey area after excluding the low-field tail, which is an artefact resulting from fast sweeping from the high to low frequencies (Supplementary Note 5)], characterised by a longer tail in the high-frequency side. This is the typical $^{75}$As NR spectrum shape[10,11,25] that includes the Knight shift caused by non-zero electron spin polarisation near $\nu = 1$. The Knight shift from the bare NR frequency $2f_{75\text{As}} = 115.809$ MHz, determined by global NR measurements (Supplementary Note 7), to a lower frequency by $2K_s$ [see Fig. 1b] is accompanied by the high-frequency tail formed by the Knight shift distribution due to variation of electron density along the growth direction of the quantum well[25]. The intensity of the NR spectrum is, however, strongly suppressed at the centre (point ii, black curve) and the right-side edge (point iii, blue curve) of the Hall bar.

## NR-intensity mapping

First, we examine the spatial variation of the NR intensity near the critical currents of QH breakdown at around $\nu = 1$ (Supplementary Note 8). The NR intensity was mapped at $B = 7$ T (for the detail procedure see Methods section) using the scanning NR metrology illustrated in Fig. 2a. After positioning the tip with RF $E$-field irradiation at the off-resonance frequency ($f_{\text{off}} = 101.050$ MHz), the frequency was switched to the on-resonance frequency of $^{75}$As nuclei ($f_{\text{on}} = 101.265$ MHz for $B = 7$ T). This was accomplished by superimposing frequency modulation $f_{\text{mod}} = \pm 70$ kHz on $f_{\text{on}}$, which integrated the NR intensity over the frequency range of the broad NR spectrum. We then recorded the variation in $V_x$ ($\Delta V_x$) due to the local NR. Note that the subtraction of the off-resonance $V_x$ would cancel a possible influence of RF $E$-field heating on the $\Delta V_x$. After setting the frequency to the off-resonance value of $f_{\text{off}}$ to recover $V_x$, the tip was moved to the next position. To get the NR intensity sufficient for imaging, the RF power in this measurement was set to 3 dBm in the nonlinear regime (Supplementary Note 9). The resultant intensity images taken as an area (dashed square in Fig. 2b) are displayed in Fig. 2c–e, demonstrating distinct patterns transformed with $\nu$. At $\nu = 1.10$ (Fig. 2c), we can see a line pattern extending over a distance of 25 μm in the $x$-direction along a Hall bar mesa edge marked by dash lines with the notation H,

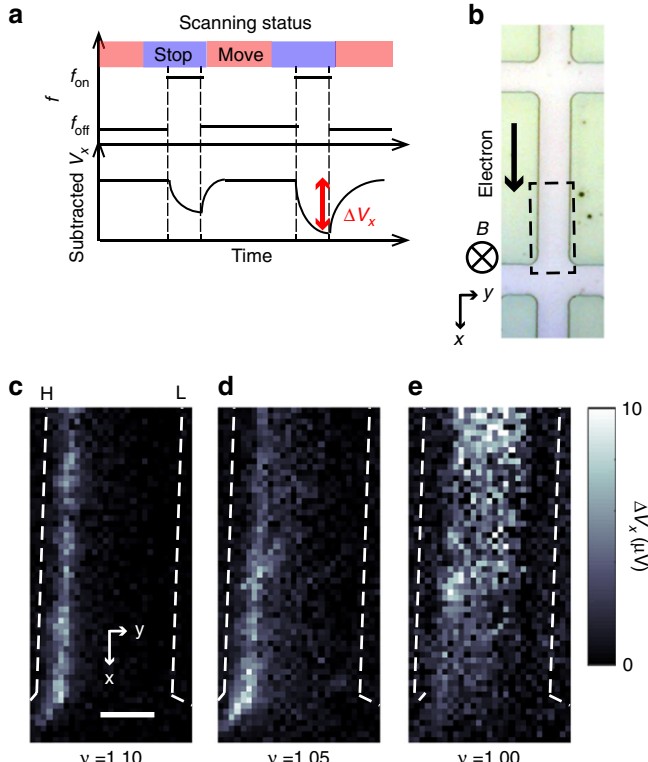

**Fig. 2** NR-intensity mapping. **a** Timing diagram of scanning NR measurements. After positioning the tip to a measuring point, the RF electric field is switched from an off-resonance frequency $f_{\text{off}}$ to an on-resonance frequency $f_{\text{on}}$. Nuclear depolarisation due to NR is detected as a reduction ($\Delta V_x$) in the subtracted $V_x$ (defined in the same manner as Fig. 1c). **b** Optical micrograph of the Hall bar. The directions of the $B$-field and electron drift are indicated. **c–e** NR-intensity images (for detailed procedures see Methods section) taken from sample A at $B = 7$ T near the critical current of QH breakdown at different values of $\nu$ [indicated by crosses in $I_x$–$\nu$ plots of $V_x$ (Supplementary Fig. 6)] in the area marked by the broken square in the Hall bar (**b**). The dash lines with notation H and L indicate the Hall bar mesa edges of the higher and lower chemical potential sides, respectively. The scale bar = 4 μm

which corresponds to the side with the higher chemical potential ($\mu_{\text{chem}}$) across the $y$-direction of the Hall bar. This $\mu_{\text{chem}}$ dependency was confirmed by an experimental result (Supplementary Note 10), showing that the pattern was relocated along the other side of the mesa edge when reversing the direction of the electron drift, and hence, the $\mu_{\text{chem}}$ direction. We then reduced $\nu$ and examined how the observed pattern changed. At $\nu = 1.05$ (Fig. 2d), the pattern slightly shifts and spreads away from the mesa edge. At $\nu = 1.00$ (Fig. 2e), the pattern eventually covers the interior of the Hall bar.

The observed $\nu$-dependent patterns can be explained as regions where DNP was driven by the inter-LL scattering[6] between the oblique LL0↑ and LL0↓ in the incompressible QH strip. The incompressible strip in the QH regime is theoretically[26] and experimentally[21] known to exist along both edges of 2DEG near both the mesa edges above integer $\nu$, and it shifts and spreads to the bulk of 2DEG with decreasing $\nu$. Such a $\nu$-dependent distribution of the incompressible strip was confirmed by the image of the Hall potential in the QH breakdown regime[27], which dominantly developed an incompressible strip along the side of the higher $\mu_{\text{chem}}$ edge. In this scenario, the RD sensitivity could be also spatially varied, and hence, it could cause the $\nu$-dependent

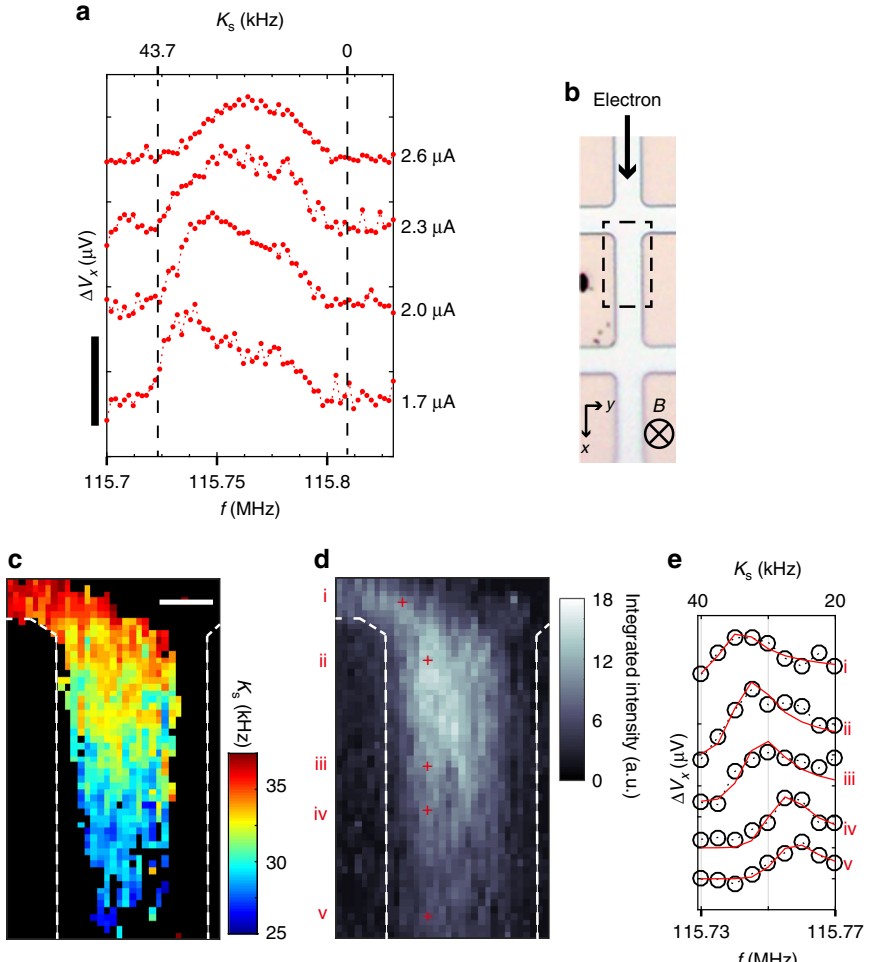

**Fig. 3** NR-spectroscopic mapping. **a** Global nuclear resonance spectra captured at QH breakdown currents (indicated in the figure) at $v = 1.02$ and $B = 8$ T for sample B. The spectrum peaks show the Knight shift scaled by the upper horizontal axes within the minimum and maximum shifts $K_s = 0-43.7$ kHz (half the value of $2K_s$), which were determined by fitting the reference data taken by pump-and-probe RD-NR measurements under unpolarised and fully polarised QH states (Supplementary Note 7). The curves are vertically offset for clarity. The vertical scale bar = 10 μV. **b** Optical micrograph of the Hall bar of sample B. The dashed square indicates a scanning area for NR spectroscopic mapping. The directions of the $B$-field and electron drift are also depicted. **c**, **d** NR Knight-shift (**c**) and intensity (**d**) images obtained by NR spectroscopic mapping (see Methods section); scale bar = 4 μm. The local $K_s$ (**c**) is plotted when the intensity (**d**) is sufficiently large to deduce $K_s$ at a cutoff intensity of 5.8 arb. units; otherwise, it is filled in with black. **e** Local NR spectra (circles) with fitting curves (red curves) at representative points marked by crosses in **d**, as indicated by i–v. The upper horizontal axis is scaled to half the value of $2K_s$. Spectra taken at different points are offset vertically for clarity

patterns; however, such an artefact was ruled out by additional NR-intensity mapping using a pump-and-probe technique (Supplementary Note 11). The observed patterns therefore indicate the $v$-dependent distribution of the DNP.

**NR spectroscopy**. We then addressed the microscopic observation of electron spin polarisation by measuring the Knight shift proportional to the spin polarisation. Prior to local measurements, we inspected the global NR spectra by applying the RF $E$-field from the back gate[24] to the entire Hall bar. Figure 3a shows the spectra obtained by the steady-state RD-NR measurements (see Methods section) at $v = 1.02$ in the QH breakdown regime. The spectrum captured at $I_x = 1.7$ μA shows an apparent peak near the NR frequency with a maximum Knight shift $K_s = 43.7$ kHz, which was determined by fitting the NR spectra (see Methods section) obtained at fully spin-polarised and spin-unpolarised QH states (Supplementary Note 7). Note that the tail at the high-frequency side of the peak is caused by variation in electron density along the growth direction of the quantum well[25]. With increasing $I_x$, the spectrum peak is

broadened with an accompanying tail on the lower-frequency side. We interpret the cause of this spectral line as the spatial variation in the electron spin polarisation formed in the QH breakdown regime.

**NR-spectroscopic mapping**. To substantiate this interpretation, we performed NR-spectroscopic mapping (see Methods section) at $I_x = 2.6$ μA within the area marked in Fig. 3b. By fitting the obtained spectra, we attained both the Knight shift and intensity images, as shown in Fig. 3c, d, respectively. Figure 3e depicts the representative spectra (circles) with the fitting curves (red curve) at spatial points marked by crosses along the expected path of the electron drift within the bulk pattern (such as Fig. 3d) observed in the intensity image. The spectrum peak shift, from 115.74 to 115.76 MHz between point i on the electron injection side (the upper-left voltage probe) and point v on the lower side of the Hall bar, demonstrates that $K_s$ decreases spatially from 35.5 to 27.5 kHz. This trend is strikingly obvious on the $K_s$ image (Fig. 3c), indicating a spatial reduction in the electron spin polarisation $P_e$.

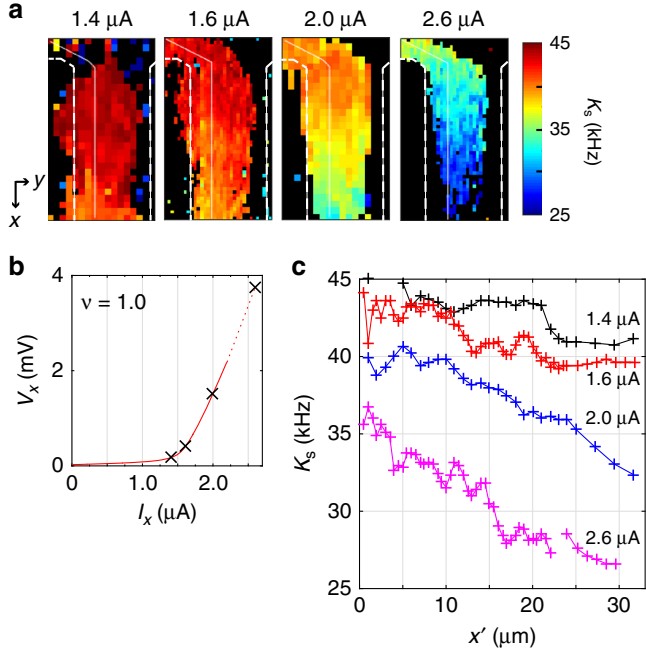

**Fig. 4** Current evolution of electron spin polarisation. **a** Knight-shift images obtained at $v = 1.02$ at different currents: $I_x = 1.4$, 1.6, 2.0, and 2.6 µA. **b** $V_x$–$I_x$ curve at $v = 1.02$. Crosses indicate the measurement points for the Knight-shift images. **c** Current dependence of $K_s$ line profiles taken along the solid white line in **a** as a function of position from the electron injection side

To examine the impact of $I_x$ on the observed spatial variation in $P_e$, we further took $K_s$ images in the same area of Fig. 3c at lower values of $I_x$, down near to the critical current ($I_c \simeq 1.5$ µA) of QH breakdown. Figure 4a shows the resulting images taken in the range 1.4 µA $\leq I_x \leq$ 2.6 µA (see $V_x$–$I_x$ curve shown in Fig. 4b). The spatial variation in $K_s$ observed along the $x$-direction at $I_x = 2.6$ µA is already substantial at $I_x = 2.0$ µA, and weakens near the critical current at $I_x = 1.4$ and 1.6 µA. This current dependence is further evident in the line profiles (shown in Fig. 4c) as a function of $x'$ along the path of electron drift (white line from the upper-left to bottom, marked in Fig. 4a). At $I_x = 1.4$ µA, the profile in most of the region exhibits almost constant $K_s$, which reaches a value close to the maximum $K_s$ (43.7 kHz) over the region $x' = 5$–20 µm. This tendency is maintained at slightly larger current $I_x = 1.6$ µA. However, further increasing the current to $I_x = 2.0$ and 2.6 µA modifies the spatial variation, which then monotonically and significantly decreases over the entire range. The maximum reduction in $K_s$ is observed at $I_x = 2.6$ µA, $\Delta K_s \simeq 9.2$ kHz, corresponding to the variation $\Delta P_e \simeq 21\%$.

## Discussion

These results provide compelling direct evidence that the spin polarisation, which is relatively homogeneous near the critical current, decreases spatially above it as the electron drifts along the $x$-direction. The spatial gradient in $P_e$ evolves with $I_x$, for example, by a factor of 1.6 when increasing $I_x$ from 2.0 to 2.6 µA, thus broadening the $K_s$ distribution. This trend is consistent with our interpretation that the tail on the low-frequency side observed at large values of $I_x$ in the global NR spectra (Fig. 3a), is due to the spatial variation of $K_s$. The observed spatial variation of electron depolarisation can be explained by a reduction in the population in the lowest LL0↑ due to the excitation of a nonequilibrium electron from LL0↑ to the upper LL0↓,

hence elevating the electron temperature, $T_e$. Such variation in $T_e$ along the direction of the electron drift has been observed in a Hall bar using a scanning cyclotron emission microscope[28], and it can be explained by the bootstrap-type electron-heating model, which has satisfactorily explained transport data previously obtained in the QH breakdown regime[29]. Near the critical current, bootstrap-type electron heating requires a certain distance from the electron injection side to be significant so that the spatial variation in $T_e$ is suppressed, at least in the measurement region over 30 µm, from the voltage probe region where cold electrons are injected (Supplementary Note 13). At a higher Hall field (larger $I_x$), however, the enhanced electron heating can affect $T_e$ and increase its spatial gradient.

We applied our NR scanning imaging technique to a hyperfine-coupled quantum system and successfully visualised the spatial distribution of DNP and electron spin polarisation. The results demonstrated the microscopic origin of the QH-breakdown-related phenomena: firstly, the spatial distribution of the DNP is correlated with the locus of an incompressible QH strip; secondly, the fully spin-polarised QH electron state is depopulated by electron heating, resulting in the non-uniform distribution of the electron spin polarization along the electron drift path. The latter, in particularly, provides essential microscopic information that complements our understanding of the QH phenomena as witnessed through conventional global measurements. This powerful technique can be utilised for direct microscopic examination of various hyperfine-coupled quantum systems, such as the QH Skyrmions[4,8] chiral QH edge states[12], helical nuclear magnetism[30], and spin-helical topological surface states[31]. Moreover, our methodology can be extended to other unique nuclear spin-related measurements, such as microscopic nuclear relaxation measurements that can directly probe local electron spin modes as in QH ferromagnet[3,4,8] and canted antiferromagnet[11].

## Methods

**Local NR excitation**. Local NR excitation was carried out by applying an RF voltage to a metallic tip, i.e., a high-impedance source that predominantly generates an RF electric field near the tip. The induced RF electric field couples with the electric quadrupole moments of $^{75}$As nuclear spins (Supplementary Note 1). The sample tip distance was maintained at 5–10 nm using feedback from a tuning fork-based atomic-force microscope. To minimise the influence of a static tip potential on the electron system within the quantum well, a DC bias voltage of 0.2 V was simultaneously added to the tip. This compensated for the potential mismatch between the tip and the surface (Supplementary Note 2).

**Resistive detection of NR**. When the nuclear spins are polarised $\langle I_z \rangle$, surrounding electrons experience an additional magnetic field $B_N = A_H \langle I_z \rangle / (g^* \mu_B)$ owing to the hyperfine interaction (Supplementary Note 3), where $A_H$ is the hyperfine coupling constant, $g^*$ is the effective electron $g$-factor and $\mu_B$ is the Bohr magneton. This so-called Overhauser field modifies the electronic Zeeman energy $E_Z \propto (B + B_N)$. In the QH breakdown regime, the DNP can be driven by inter-LL scattering between the oblique LL0↑ and LL0↓, and it induces a negative $B_N$-field through negative $g^*$ of GaAs[6], eventually reducing $E_Z$. This decreases the separation between LL0↑ and LL0↓, and therefore, enhances the inter-LL scattering causing the non-zero longitudinal resistance in the QH breakdown regime. Thus, the NR-induced nuclear spin depolarisation decreases the $V_x$ at constant $I_x$, leading to the NR spectrum, for example, as shown in Fig. 1c. Further detail discussion appears in Supplementary Note 3.

**Steady-state RD-NR measurements**. To obtain the exact NR spectral shape for the local spectra (Fig. 1c), after pumping the DNP at the off-resonance frequency $f_{off} = 115.90$ MHz for the time duration of more than 15 s (Supplementary Note 6), we performed the NR at an on-resonance frequency for the NR saturation time, 12–15 s, until the steady state (Supplementary Note 6) was obtained. Eventually, we measured the variation in $V_x$ ($\Delta V_x$). By repeating these procedures in the range of $f_{on} = 115.67$–115.83 MHz, the steady-state NR spectrum was obtained (Supplementary Note 5). During the entire sequence, all other conditions of $I_x$, $v$, and $B$ were maintained constant. The global NR spectra (Fig. 3a) was also obtained by the steady-state measurements at each $I_x$ indicated in the figure.

**NR-intensity mapping**. As shown in Fig. 2a, after positioning the tip to a measuring point, the frequency of the RF $E$-field was switched from $f_{off} = 101.050$ MHz to $f_{on} = 101.265$ MHz, and additionally modulated by $f_{mod} = \pm 70$ kHz, which integrates the intensity over the frequency range of the broad NR spectrum. The NR was performed at RF power $P_{RF} = 3$ dBm for $t_{on} = 12$–15 s until the steady state is reached (Supplementary Note 6), and thereby, the NR-induced reduction in $V_x$ ($\Delta V_x$) was measured. The frequency was then switched back to $f_{off}$ and was kept for $t_{off} > 15$ s to recover the DNP (Supplementary Note 6). By mapping the $\Delta V_x$, the NR-intensity images were obtained.

**Evaluation of Knight shift by fitting NR spectra**. The NR spectra $I(f)$ at $K_s = \alpha |\psi_{NR}(z)|^2$, with parameter $\alpha$ representing the size of Knight shift, were fitted using a Gaussian function[10]:

$$I(f) = A \int_{-w/2}^{w/2} |\psi_{read}(z)|^2 \exp(\frac{-(f - f_0 + \alpha|\psi_{NR}(z)|^2)^2}{2\Gamma^2})dz \qquad (1)$$

where both 2DEG wave functions along the direction ($z$) of the growth of the quantum well for reading [$\psi_{read}(z)$] and NR [$\psi_{NR}(z)$] were assumed to be $\cos(\frac{\pi z}{w})$ for the 2DEG width $w = 20$ nm. By fitting the reference data obtained at the $\nu = 2$ QH state (Supplementary Note 7), the $2f_{75As}$ at $K_s = 0$ and width of the spectral line were determined as $f_0 = 115.809$ kHz and $\Gamma = 4.63$ kHz, respectively. The maximum Knight shift corresponding to twice the fundamental Knight shift ($2K_s$) was determined to be 87.3 kHz by fitting the reference data obtained at the $\nu = 1$ QH state (Supplementary Note 7). To avoid any confusion with the Knight-shift value, we use half the value of $2K_s$ throughout this article; for example, when the Knight shift is maximum, $K_s = 87.4/2 = 43.7$ kHz.

**NR-spectroscopic mapping**. NR-spectroscopic data were constructed by mapping $\Delta V_x$ at the arbitrary $f_{on}$ using the procedure shown in Fig. 2a (see also Supplementary Note 12 for details). We extracted the intensity at fixed spatial points from the dataset of $\Delta V_x$ images for different values of $f_{on}$ in the range 115.730–115.770 MHz to reconstruct local NR spectra shown in Fig. 3e.

**Data availability**. The data that support the findings of this study are available from the corresponding authors on request.

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

## Acknowledgements

We thank K. Muraki and NTT for supplying high-quality wafers, T. Taninaka and S. Shirai for the sample preparation, and M. Kawamura, T. Machida, N. Kumda, M.H. Fauzi, J.F. Ribeiro, G. Yusa, and J.N. Moore for the helpful discussions. K.H. acknowledges JSPS for financial support (KAKENHI 26390006, 17H02728). Y.H. acknowledges support from JSPS (KAKENHI 15H0586, 15K217270, and 26287059) and Tohoku University, WPI-AIMR. K.H. and Y.H. thank Tohoku University's GP-Spin program for their support.

## Author contributions

K.H. performed the experiments and wrote the paper with the help of T.T. and K.S., and with advice of Y.H. K.H. created the entire experimental design and conducted the data analysis.

## Additional information

**Competing interests:** The authors declare no competing interests.

