## [Peer Review File · Nature Communications]

Reviewers' comments:

Reviewer #1 (Remarks to the Author):

The authors investigate a quantum Hall system at filling factors close to 1 (nearly spin polarized system) with the established method of resistive detection of nuclear resonance. In these experiments, electrically excited electrons (breakdown of quantum Hall effect) create dynamically a nuclear spin polarization, which influences the Zeeman splitting of the electrons via the Overhauser field and therefore the magnetoresistance. New is the use of a scanning probe for spatially resolved NMR measurements. The tip couples locally to the electrically active quadrupole transition and the measured variation in the longitudinal resistance gives information about:

a) The current induced nuclear spin polarization which show strong variation in the spatial distribution if the filling factor is slightly changed
and

b) the inhomogeneous spin polarization of the 2-dimensional electron gas under the condition of electrical breakdown of QHE (analysis of Knight shift) .

The most important results are summarized in Fig. 2 where the reduction of the Knight shift in current direction visualizes the increase of the electron temperature and consequently the electron spin polarization at filling factor 1 under QHE breakdown condition. In addition, the results in Fig. 1 confirm nicely the correct picture for the filling factor dependent current distribution in quantum Hall systems under nonohmic conditions. If the filling factor changes from an integer value (Fig. 1f) to a slightly higher filling factor (Fig 1d, 1e), a concentration of the current flow close to the incompressible strip of filling factor 1 at the side for the higher electrochemical potential is observed.

Even if most of the presented results are not surprising, the experimental technique is new (and demanding) so that I recommend publication. However, the authors should address the following questions and improve the manuscript:

The detection scheme relies on a change in the magnetoresistance if the microwave excites NR.

The detailed mechanism NR-change in Zeeman energy-change in resistance is not described.

The scanning tip is invasive even if the work function difference of about 0.2 Volt is compensated since the source-drain voltage is not negligible. This may change the sensitivity of the detection signal especially for the data in Fig. 1e and 1d (local distortion of the filling factor). Can one rule out microwave induced heating if the microwave frequency is changed from off- to on-resonance? Such a heating may produce similar results as shown in Fig. 1.

The title “integrated NR intensity mapping” gives the impression, that quantitative measurements

over the whole area of a sample is possible. This is not true if the current flow is not homogeneous as discussed in Fig S5 . The NR cannot be measured if the position of the detecting current is not correlated with the position of the dynamically polarized nuclear spin (e.g. nuclear spin diffusion?).

How reliable is the claim, that the spatial resolution of the NR experiment is better than 500 μm ? The narrow feature in Fig S6 probably originates from a strongly nonlinear change in the resistance with position.

Many different time intervals (13 s, 23 s, 39 s and 12 s) are used in the measurement scheme which confuses the reader. Nothing is mentioned about other time constants like the nuclear relaxation time which varies drastically in the neighborhood of filling factor 1 (Skyrmions?).

Reviewer #2 (Remarks to the Author):

Titel:

Scanning nuclear resonance imaging of a hyperfine coupled quantum Hall system

Hashimoto et al

I think the aim of the paper is to present an imaging technique that plots changes in the resistivity of a sample due to nuclear resonance excited by a scanning tip that excites the resonance locally. The aim is to learn about the local electronic structure of (two-dimensional) Quantum Hall systems.

I am not an expert in the field, but I think the resistivity-detected nuclear resonance is established, but not in combination with the special excitation using a tip.

However, for an experimental physicist who is not new to the physical principles involved it is close to impossible to understand what is going on, even after various reboots in reading. If other reviewers think the manuscript should be published, I demand that changes along the very long list below are made. Maybe it will be possible then to get an interesting paper, but - honestly - I doubt that since the trustable new insight, apart from giving something new, is rather limited.

A number of examples of concrete shortcomings is listed below.

Abstract

Sentence 1: Why “nuclear polarization” and “electronic spin polarization”, one wonders since leaving out the spin for the nuclei is expected to mean something (of course, it is the electrons that often have polarizations that are not only spin - so is it intentional? No.)

Sentence 2: In regular NMR and EPR experiments the hyperfine coupling is studied routinely, also the spatial variation. So maybe only for 2DEG it is not done. Then it is not said whether “spin polarization” refers to the electrons or nuclei. It could be both, probably?

Sentence 4: Now it comes that both spin polarizations are spatially modulated, however, at least for the regular NMR language the Knight shift (as a relative shift) is proportional to the susceptibility, not the polarization (e.g., no Knight shift in a magnetic material).

Regular Text

Sentence 1: “strong magnetic field”, yes I know, one does that in high magnetic fields, but, later in the text I was looking for a field value and searched for “magnetic”. One hit at the bottom of page 8, but there it is used in “magnetic quantum number $|m\rangle$ ” - of course that is the typical labelling for a spin state with magnetic quantum number “m”; no value of the field given in the regular text. That is unacceptable.

Is it only the contact term of the hyperfine interaction? It could be mentioned why?

Too many citations without saying anything, 1 to 9. The reader will not look up all papers and see which spin states there are. This comes often. Citations as necessary piece of information should be used rarely.

First sentence page 2: “allows us” this should probably be “allows the”

Second sentence: What is “linearly proportional”, both, or a new mathematical dependence?

Next paragraph: Again, which “spin polarization”? Both?

“which afforded spatial information on DNP”? do they mean “provided”

Outgoing paragraph on p.2: Used but not defined what is meant by “NR intensity”; what is that thing?

The Knight shift is again proportional (not linearly prop.) to the polarization which is χB_0 .

Results

Scanning...

The 4th and 5th line are not acceptable: NR frequency is not defined, then, “quadrupolar characteristics” of nuclear spin - this is not physics.

The total spin $|I| > 1/2$: spin is not negative, so what does that mean?

As - is not a variable;

A few words on the energy levels and excitation are expected, also a word why it really works (bandwidth of excitation, etc)

Integrated...

“spreading in the frequency due to the Knight shift” sounds not good, but also leaves open if the Knight shift distribution is what is important.

2nd paragraph: suddenly DNP shows up, a complicated thing that was never explained (how it

works here).

.... I stop here.

Reviewer #3 (Remarks to the Author):

In this paper, the authors investigate the non-uniform distribution of the electron polarization in the quantum Hall (QH) system, using locally excited, resistively detected nuclear resonance. The resonance was excited by applying a localized radio-frequency electric field via AFM tip; by measuring the Knight shift of the resonance peak, quantitative information about the spatial distribution of the electron polarization was extracted. The authors investigated several QH regimes, including the breakdown regime, where the electron spin polarization is strongly non-homogeneous along the sample. As a result, the authors, firstly, demonstrated the potential of a new microscopic imaging technique, and, secondly, obtained an interesting direct information about the microscopic structure of the QH system as a function of the current. In my opinion, the results of this paper are of interest for a broad community of physicists working in different areas, and have a good chance to attract attention of the audience of Nature Communications. Thus, in my opinion, the paper can be published, although some revisions would be desirable, see the list below.

- 1) In the abstract, the authors claim that their results "reveal the microscopic origin of the nonequilibrium QH phenomena". While, indeed, the work directly demonstrates the microscopic distribution of the spin polarization, it remains somewhat unclear how do the results reveal "the microscopic origin", and which specific "non-equilibrium QH phenomena" are implied (the breakdown regime?) I think that clarifying this point would be useful for the future readers.
- 2) The resistive readout of the nuclear resonance is described by two very generic sentences in the "Methods" section. Given the importance of the readout method, I think it deserves a somewhat more extensive discussion, perhaps in the Supplemental Materials.
- 3) It remains somewhat unclear whether the measurements are performed in the linear response regime. For instance, is the NR signal proportional to the amplitude of the rf driving E-field?
- 4) Some curves show additional features (e.g. Fig. 2a, second curve from the top with an extra peak to the left of the main one, or the bottom curve of Fig. 2a and the curve in Fig. S8, showing noticeable extra peaks to the right of the main one). Is anything known about the possible origin of these peaks, or are they just noise features?

Summarizing, I think the work presents new results, interesting for a broad audience of Nature Communications, and the technique demonstrated by the authors has much potential. I think the work deserves publication in Nature Communications, although some revisions might help the future readers.

REPLY TO REVIEWER #1:

We thank the Reviewer for his/her positive recommendation. We provide below point-by-point replies, highlighted in blue.

The authors investigate a quantum Hall system at filling factors close to 1 (nearly spin polarized system) with the established method of resistive detection of nuclear resonance. In these experiments, electrically excited electrons (breakdown of quantum Hall effect) create dynamically a nuclear spin polarization, which influences the Zeeman splitting of the electrons via the Overhauser field and therefore the magnetoresistance. New is the use of a scanning probe for spatially resolved NMR measurements. The tip couples locally to the electrically active quadrupole transition and the measured variation in the longitudinal resistance gives information about:

- a) The current induced nuclear spin polarization which show strong variation in the spatial distribution if the filling factor is slightly changed and
- b) the inhomogeneous spin polarization of the 2-dimensional electron gas under the condition of electrical breakdown of QHE (analysis of Knight shift).

The most important results are summarized in Fig. 2 where the reduction of the Knight shift in current direction visualizes the increase of the electron temperature and consequently the electron spin polarization at filling factor 1 under QHE breakdown condition. In addition, the results in Fig. 1 confirm nicely the correct picture for the filling factor dependent current distribution in quantum Hall systems under nonohmic conditions. If the filling factor changes from an integer value (Fig. 1f) to a slightly higher filling factor (Fig 1d, 1e), a concentration of the current flow close to the incompressible strip of filling factor 1 at the side for the higher electrochemical potential is observed. Even if most of the presented results are not surprising, the experimental technique is new (and demanding) so that I recommend publication. However, the authors should address the following questions and improve the manuscript:

We appreciate the positive judgment of the Reviewer. To address the questions raised by the Reviewer, we added descriptions and figures to the manuscript and Supplementary materials, marked in red, such that our paper has now been significantly improved.

The detection scheme relies on a change in the magnetoresistance if the microwave excites NR. The detailed mechanism NR-change in Zeeman energy-change in resistance is not described.

The general explanation of the resistive detection has been moved from Methods to

the first paragraph of the subsection, 'Local NR'. Further, an extensive discussion has been added in Methods.

The scanning tip is invasive even if the work function difference of about 0.2 Volt is compensated since the source-drain voltage is not negligible. This may change the sensitivity of the detection signal especially for the data in Fig. 1e and 1d (local distortion of the filling factor).

The source-drain voltage used for the intensity mapping was below 40 mV, which could be an additional source of local potential mismatch between the tip and the sample surface. We tested the influence of the potential mismatch on the edge pattern [as shown in the former Fig. 1d (new Fig. 2d)] by changing the DC tip voltage V_{tip} instead of the source-drain voltage. Figure S1b (added to Supplementary materials) shows line mappings of the nuclear resonance intensity captured at different V_{tip} along the same line across a Hall bar. The peak pattern near $y = 2.5 \mu\text{m}$ persists in the V_{tip} range of ± 0.4 V with respect to the $V_{\text{tip}} = 0.2$ V used for compensation of the work function difference such that the corresponding range of the potential mismatch does not affect the resistive detection sensitivity. We thus concluded that the source-drain voltage < 40 mV used in our measurements did not affect the detection sensitivity.

Can one rule out microwave induced heating if the microwave frequency is changed from off- to on-resonance? Such a heating may produce similar results as shown in Fig. 1.

Since the same microwave power was used for both the off- and on-resonance procedures, a possible heating effect caused by microwave was basically cancelled by substituting the off-resonance V_x from on-resonance V_x . This explanation has been added in the subsection 'NR spectral intensity mapping'.

Moreover, we determined the on-resonance frequency by identifying ^{75}As nuclear resonance in the continuous wave spectra (new Fig. 1c), which shows a trend consistent with the edge patterns (new Fig. 2c, 2d). This clearly indicates that the pattern observed in the intensity mapping reflects V_x reduction associated with the nuclear resonance, rather than microwave heating.

The title "integrated NR intensity mapping" gives the impression, that quantitative measurements over the whole area of a sample is possible. This is not true if the current flow is not homogeneous as discussed in Fig S5. The NR cannot be measured if the position of the detecting current is not correlated with the position of the dynamically

polarized nuclear spin (e.g. nuclear spin diffusion?).

Indeed, for the 'integrated NR intensity mapping' we cannot exclude the influence of inhomogeneous detection sensitivity particularly at a high filling factor $\nu = 1.10$. The combination of 'integrated' with 'intensity mapping' perhaps gives the impression of quantitative measurement. The subsection title has been replaced simply by 'NR-intensity mapping'.

One candidate for the quantitative scanning measurements is to implement pump-and-probe technique based on ν tuning via the back gate (Supplementary S8). For instance, the bottom curve in Fig. S8 was obtained by pumping the DNP at $\nu = 1.13$ while sensing the NR intensity at $\nu = 1.06$, having relatively homogeneous detection sensitivity. The NR intensity that increased near the Hall-bar edge was strongly suppressed in the Hall-bar interior. This suggests less contribution of the nuclear spin diffusion. Further extensive measurements such as corresponding 2D mapping are expected, but beyond the scope of this article.

How reliable is the claim, that the spatial resolution of the NR experiment is better than $500 \mu\text{m}$? The narrow feature in Fig S6 probably originates from a strongly nonlinear change in the resistance with position.

We completely agree with the Referee's comment, and have removed both the sentence and figure from the manuscript and Supplementary materials.

Many different time intervals (13 s, 23 s, 39 s and 12 s) are used in the measurement scheme which confuses the reader. Nothing is mentioned about other time constants like the nuclear relaxation time which varies drastically in the neighborhood of filling factor 1 (Skyrmions?).

Our time constant of NR-induced depolarisation relies on the steady state determined by several rates such as the nuclear relaxation, NR transition (depolarisation), DNP pumping, and nuclear diffusion rates. Indeed, quantum-Hall Skyrmions can strongly enhance the nuclear relaxation rate, but they are less effective in the QHE breakdown regime used in this work. We determined the appropriate time intervals by directly measuring the effective time constant instead of deducing each of the time constants. As shown in Supplementary S4, we measured the time evolution of V_x by applying the RF electric (E) field at a point where the DNP arises in the interior of the Hall bar at $\nu \sim 1.0$. The determined NR saturation time (t_{NR}) of the NR-induced nuclear spin depolarisation and the recovery time (t_{DNP}) of the DNP were typically $t_{\text{NR}} \sim 12 \text{ s}$ and $t_{\text{DNP}} \sim 15 \text{ s}$. The time intervals satisfying t_{NR} and t_{DNP} are used for the NR procedures of the

on (t_{on}) and off (t_{off}) resonance within the range of $t_{\text{on}} = 12\text{-}15$ s and $t_{\text{off}} > 15$ s. To avoid confusing the readers, these ranges of time intervals have been used instead of each time interval. Accordingly, 'Pump-and-probe measurements of global NR spectra' formerly present under Methods and the local NR measurement for 'NR-intensity mapping', which both used the time intervals within the above-mentioned ranges, were renamed as 'steady-state NR measurement'.

REPLY TO REVIEWER #2:

We thank the Reviewer for his/her helpful questions and remarks. We have addressed all the comments as provided below in blue. This revision has indeed improved the presentation of our works attractive for a broad community of physicists working in different areas.

I think the aim of the paper is to present an imaging technique that plots changes in the resistivity of a sample due to nuclear resonance excited by a scanning tip that excites the resonance locally. The aim is to learn about the local electronic structure of (two-dimensional) Quantum Hall systems.

I am not an expert in the field, but I think the resistivity-detected nuclear resonance is established, but not in combination with the special excitation using a tip. However, for an experimental physicist who is not new to the physical principles involved it is close to impossible to understand what is going on, even after various reboots in reading. If other reviewers think the manuscript should be published, I demand that changes along the very long list below are made. Maybe it will be possible then to get an interesting paper, but - honestly - I doubt that since the trustable new insight, apart from giving something new, is rather limited.

According to the comments, we have fully or partly rewritten some subsections in text as well as added several figures to both the manuscript and Supplementary materials. The revised paper indeed presents the abilities of our microscopic imaging technique unambiguously, providing new insights of the microscopic distribution of both nuclear and electron spin polarizations in a hyperfine-coupled quantum Hall system. Firstly, the spatial distribution of the DNP strongly links the local QH electron states. Secondly, the fully spin-polarised QH electron state is depopulated by the electron heating, resulting into the non-uniform distribution of the electron spin polarization along the electron drift path.

Abstract

Sentence 1: Why “nuclear polarization” and “electronic spin polarization”, one wonders since leaving out the spin for the nuclei is expected to mean something (of course, it is the electrons that often have polarizations that are not only spin - so is it intentional? No.)

According to the suggestion, we changed ‘nuclear polarisation’ to ‘nuclear spin polarisation’.

Sentence 2: In regular NMR and EPR experiments the hyperfine coupling is studied routinely, also the spatial variation. So maybe only for 2DEG it is not done. Then it is not said whether "spin polarization" refers to the electrons or nuclei. It could be both, probably?

To our knowledge, conventional imaging techniques such as MRI cannot resolve structures of a few microns of the spin polarization, which we succeeded in imaging in this work. Nevertheless, here we refer to both spin polarization; the phrasing has been updated accordingly.

Sentence 4: Now it comes that both spin polarizations are spatially modulated, however, at least for the regular NMR language the Knight shift (as a relative shift) is proportional to the susceptibility, not the polarization (e.g., no Knight shift in a magnetic material). The Knight shift is again proportional (not linearly prop.) to the polarization which is χB_0 .

Our definition of the Knight shift is different from the usual one. Generally, the Knight shift is defined as the relative shift, i.e., the NMR line shift divided by the reference line position proportional to the external magnetic field. This definition is of use when the NMR line shift induced by the conduction-electron magnetisation is proportional to the external magnetic field. The corresponding Knight shift is thus defined by the NMR line shift divided by the reference line position, providing the proportional coefficient, namely susceptibility.

On the other hand, for the quantum Hall system, the NMR line shift is used to directly measure the electron spin polarisation (magnetisation), which relies on either even or odd Landau level filling factors, and hence, is not proportional to the external magnetic field. Thus, we use the absolute value of the NMR line shift proportional to electron spin polarisation. This definition has been widely used for direct observation of various quantum-Hall spin polarization via optically-pumped NMR [1,2], resistively-detected NMR [3,4]. Brief explanations concerning the difference in the definition have been added to the end of the first paragraph of the main text.

[1] Tycko, R., et al. "Electronic states in gallium arsenide quantum wells probed by optically pumped NMR." *Science* 268.5216 (1995): 1460.

[2] Kuzma, N. N., et al. "Ultraslow electron spin dynamics in GaAs quantum wells probed by optically pumped NMR." *Science* 281.5377 (1998): 686-690.

[3] Tiemann, L., et al. "Unraveling the spin polarization of the $\nu = 5/2$ fractional quantum Hall state." *Science* 335.6070 (2012): 828-831.

[4] Kumada, N., Muraki, K., and Hirayama, Y. "NMR evidence for spin canting in a bilayer $\nu = 2$ quantum Hall system." *Physical review letters* 99.7 (2007): 076805.

Regular Text

Sentence 1: "strong magnetic field", yes I know, one does that in high magnetic fields, but, later in the text I was looking for a field value and searched for "magnetic". One hit at the bottom of page 8, but there it is used in "magnetic quantum number $|m\rangle$ " - of course that is the typical labelling for a spin state with magnetic quantum number "m"; no value of the field given in the regular text. That is unacceptable.

Thank you for pointing out the missing description of the magnetic fields in the main text. Indeed, the magnetic fields that we used were described as ' $B = \mu_0 H$ ' only in the figure captions *without* defining ' B '. The definition and the magnetic field range we used have now been added to the end of the first paragraph in the subsection 'Local NR'.

Is it only the contact term of the hyperfine interaction? It could be mentioned why?

The two-dimensional electron gas within the GaAs/AlGaAs quantum well comprises s-type conduction band electrons, whose probability density has a sharp maximum at the nuclear sites leading to Fermi contact interactions. This explanation has been added to the first paragraph of the introduction.

Too many citations without saying anything, 1 to 9. The reader will not look up all papers and see which spin states there are. This comes often. Citations as necessary piece of information should be used rarely.

Previously, all the citations used in the introduction had been placed on the first sentence, which gave an impression of too many citations without explanations. The citations have been therefore removed from the first sentence; the citations are now used only in the subsequent sentences appropriately.

First sentence page 2: "allows us" this should probably be "allows the"

According to the Referee's suggestion, we removed 'us'.

Second sentence: What is "linearly proportional", both, or a new mathematical dependence?

'Linearly' was used to emphasize 'proportional', however, it has been removed to avoid any confusion.

Next paragraph: Again, which “spin polarization”? Both?

Here, ‘polarisation’ refers to both nuclear and electron spin polarisations; the phrasing has been updated accordingly.

“which afforded spatial information on DNP”? do they mean “provided”

We followed the Referee’s suggestion and incorporated the required revision.

Outgoing paragraph on p.2: Used but not defined what is meant by “NR intensity”; what is that thing?

As mentioned by the Referee, the ‘NR intensity’ was not defined before the overview sentences (end of the introduction): ‘In this article, we show the spatial distribution of the NR intensity and NR Knight shift’. To make this clearer, the sentence was replaced by ‘we show the spatial distribution of the DNP and electron spin polarisation’.

Results

Scanning...

The 4th and 5th line are not acceptable: NR frequency is not defined, then, “quadrupolar characteristics” of nuclear spin - this is not physics.

A few words on the energy levels and excitation are expected, also a word why it really works (bandwidth of excitation, etc)

To clarify the working of the NR technique based on the quadrupolar NR, we added the energy diagram (former Fig. S2), and new data on local ⁷⁵As NR spectra to Fig. 1 after moving the former Fig. 1b-f to new Fig. 2. The corresponding description of Fig. 1 was added as the subsection ‘Local NR’.

The total spin $|I| > 1/2$: spin is not negative, so what does that mean?

As - is not a variable;

Thank you for pointing out our typos. The symbol of absolute value has been removed and the italicized ‘As’ has been revised to normal fonts.

Integrated...

“spreading in the frequency due to the Knight shift” sounds not good, but also leaves open if the Knight shift distribution is what is important.

As mentioned, this description refers to the Knight shift distribution caused by variation of electron density along the growth direction of the quantum well [perpendicular to the two-dimensional electron gas (2DEG)]. This Knight shift distribution could be confused

with the more important in-plane distribution (parallel to the 2DEG) described later. To avoid this, we have added an explanation of the out-of-plane distribution in the second paragraph of subsection 'Local NR'.

2nd paragraph: suddenly DNP shows up, a complicated thing that was never explained (how it works here).

The original explanation of the DNP in QH breakdown (described right before Results section) was unclear. A description has been added about the DNP in the revised version.

REPLY TO REVIEWER #3:

We appreciate his/her positive comments, particularly your acknowledgement of our work, 'the work presents new results, interesting for a broad audience of *Nature Communications*, and the technique demonstrated by the authors has much potential'. We provide point-by- point replies below in blue.

In this paper, the authors investigate the non-uniform distribution of the electron polarization in the quantum Hall (QH) system, using locally excited, resistively detected nuclear resonance. The resonance was excited by applying a localized radio-frequency electric field via AFM tip; by measuring the Knight shift of the resonance peak, quantitative information about the spatial distribution of the electron polarization was extracted. The authors investigated several QH regimes, including the breakdown regime, where the electron spin polarization is strongly non-homogeneous along the sample. As a result, the authors, firstly, demonstrated the potential of a new microscopic imaging technique, and, secondly, obtained an interesting direct information about the microscopic structure of the QH system as a function of the current. In my opinion, the results of this paper are of interest for a broad community of physicists working in different areas, and have a good chance to attract attention of the audience of Nature Communications. Thus, in my opinion, the paper can be published, although some revisions would be desirable, see the list below.

We thank the Reviewer for your positive response.

1) In the abstract, the authors claim that their results "reveal the microscopic origin of the nonequilibrium QH phenomena". While, indeed, the work directly demonstrates the microscopic distribution of the spin polarization, it remains somewhat unclear how do the results reveal "the microscopic origin", and which specific "non-equilibrium QH phenomena" are implied (the breakdown regime?) I think that clarifying this point would be useful for the future readers.

The results demonstrate microscopic origin of the QH-breakdown-related phenomena: firstly, the spatial distribution of the DNP strongly links the local QH electron states; secondly, the fully spin-polarised QH electron state is depopulated by the electron heating, resulting into the non-uniform distribution of the electron spin polarization along the electron drift path. The latter, in particular, provides essential microscopic information that complements our understanding of the quantum Hall phenomena as witnessed through conventional global measurements. This conclusion was added to

the last paragraph in the main text.

2) The resistive readout of the nuclear resonance is described by two very generic sentences in the "Methods" section. Given the importance of the readout method, I think it deserves a somewhat more extensive discussion, perhaps in the Supplemental Materials.

The general mechanism of the resistive readout is now moved from Methods to the first paragraph of the subsection, 'Local NR', as suggested by Reviewer #1. Further, an extensive discussion has been added to Methods.

3) It remains somewhat unclear whether the measurements are performed in the linear response regime. For instance, is the NR signal proportional to the amplitude of the rf driving E-field?

Figure R1 shows the typical RF power dependence of the NR intensity extracted from the NR-intensity mappings measured at the centre of the hall bar in the $\nu \sim 1.0$ QH breakdown regime. The RF power used in the spectroscopy mapping [$P_{\text{RF}} = -4$ dBm (~ 0.4 mW)] is in the linear response regime below $P_{\text{RF}} \sim 0$ dBm (1 mW) where P_{RF} starts showing the nonlinear dependence. On the other hand, the intensity mapping conducted at a larger power [$P_{\text{RF}} = 3$ dBm (~ 2.0 mW)] is in the nonlinear response regime; however, the observed trend of the ν -dependent distribution in the NR intensity is also confirmed in the linear response regime at $P_{\text{RF}} = -10$ dBm (0.1 mW) as shown in the new Fig. 1c. Thus, we concluded that the ν -dependent distribution of the NR intensity displayed in new Fig. 2c–e shows intrinsic distribution of the DNP.

Fig. R1 The RF power dependence of the NR intensity, ΔV_x . The ΔV_x was extracted from the NR line mapping measured at the center of the Hall bar in the $\nu = 1$ QH breakdown regime at $B = 7$ T.

4) Some curves show additional features (e.g. Fig. 2a, second curve from the top with an extra peak to the left of the main one, or the bottom curve of Fig. 2a and the curve in Fig. S8, showing noticeable extra peaks to the right of the main one). Is anything known about the possible origin of these peaks, or are they just noise features?

We did not observe the reproducibility of the *additional features* mentioned by the Referee, and therefore conclude that they are noise features. Note that the same data was used for the bottom curve of the former Fig. 2a and the curve in the former Fig. S8, and these showed the same noise feature.

Summarizing, I think the work presents new results, interesting for a broad audience of Nature Communications, and the technique demonstrated by the authors has much potential. I think the work deserves publication in Nature Communications, although some revisions might help the future readers.

Thank you very much again for your positive response and helpful comments.

SUMMARY OF CHANGES

All changes have been marked in red to help Reviewers locate them.

-Explanations for the term 'contact interaction' and our definition of Knight shift have been added in the first paragraph of the main text.

- The energy diagram (former Fig. S2), and new data about local ^{75}As NR spectra have been added to Fig. 1 after the former Fig. 1b–f was moved to the new Fig. 2 to explain our NR technique more explicitly. The corresponding text has also been added as the subsection 'Local NR'. Accordingly, the first part of the next subsection 'NR-intensity mapping' has been rewritten.

- The physical findings are addressed in the conclusion part of the main text to answer the Reviewer's query.

-Methods explaining 'Resistive detection of NR' and 'Steady-state RD-NR measurements' (former 'Pump-and-probe measurements of global NR spectra') have been rewritten according to the Reviewers' remarks. Accordingly, a related Method 'NR-intensity mapping' has been partly rewritten.

-Supplementary Materials discussing influence of the source-drain voltage (S1), the exact NR spectrum shape (S3), and time interval for reaching the steady-state NR (S4), have been added to answer the Reviewers' questions. Further, a $V-I$ curve (S2) has been added to clarify the experimental condition used for the measurements (newly added Fig. 1c).

-Minor additional changes in the text have been made to clarify the discussions, to remove typos according to the Reviewers' remarks, or to eliminate grammatical errors.

Reviewers' comments:

Reviewer #1 (Remarks to the Author):

The authors clarified most of the open questions by adding new figures and text. For example the new figure S1b shows that the noninvasive contribution of the tip (local gate voltage) is unimportant. However some confusion exists with the sign of the signal ΔV_x . In Fig S1a the ΔV_x signal includes the sign of the measured change in the resistivity whereas in Fig S1b (and all other figures with a ΔV_x curve ?) the signal is reversed. The authors should use the expression ΔV_x in a consistent way. If I understand the results correctly, the ΔV_x signal has a negative sign since the randomization of the nuclear spin polarization leads to an INCREASE in the Zeeman energy and therefore to a reduction in the measured resistivity.

The experimentally observed NEGATIVE Overhauser field BN as a result of DNP in the breakdown regime (is this a consequence of the negative value of the electronic g-factor in GaAs and the spin flip-flop process?) should be explained in more detail.

Reviewer #2 (Remarks to the Author):

Confidential remarks to the Editor

Reviewer #3 (Remarks to the Author):

In the revised version of the paper the authors, in most part, adequately addressed my questions. I think the paper can be published, with some revisions. There are two suggestions I would like to make.

1) The graph given in Fig. R1 could be included in the Supplement. As far as I see, the response is not quite linear, and I cannot really agree with the authors that the linear regime corresponds to the RF power below 1 mW. While this non-linearity does not seem to affect the main conclusions of the paper, I think it is important to make this point clear for the audience: include the corresponding explanation into the main text, and present Fig. R1 so that the interested readers can see the linearity/nonlinearity of the measurement.

2) I am not completely sure what do the authors mean when they claim that "...the spatial distribution of the DNP strongly links the local QH electron states" (see the authors' Reply and the added sentences in the last paragraph of the main text). I think it would be useful if the authors explained this point in more detail.

I think, with the above-mentioned explanations, the manuscript can be published.

Reviewer #4 (Remarks to the Author):

I have read the revised manuscript in light of the previous round of reviews and found that the authors adequately answered the referees' requests. The revised manuscript is much more readable. I think the results are sound and should ultimately be published. However, the paper appears to have been written for PRL; for a general audience such as that of Nature Communications, which will not be familiar with resistively detected NMR, some additional explanations are needed. First of all, why write NR, and not NMR? After all, it is a *magnetic* resonant effect, no? (Or did I miss something?) In any case, the main change that is needed is to include a supplementary section describing the physics of the technique used. I suggest the authors include a section stating the relevant Hamiltonian being probed, and how readout and manipulations are done, and how this relates to current. For RD-NMR people, this is obvious, but for NCOMM readers, this is not obvious at all.

REPLY TO REVIEWER #1:

We thank the Reviewer for his/her remarks which have helped further improve our manuscript. We have provided point-by-point replies below; our responses are highlighted in blue.

The authors clarified most of the open questions by adding new figures and text. For example, the new figure S1b shows that the noninvasive contribution of the tip (local gate voltage) is unimportant.

Thank you for your comment. We appreciate that you found our revisions to be sufficient and helpful.

However some confusion exists with the sign of the signal ΔV_x . In Fig S1a the ΔV_x signal includes the sign of the measured change in the resistivity whereas in Fig S1b (and all other figures with a ΔV_x curve?) the signal is reversed. The authors should use the expression ΔV_x in a consistent way. If I understand the results correctly, the ΔV_x signal has a negative sign since the randomization of the nuclear spin polarization leads to an INCREASE in the Zeeman energy and therefore to a reduction in the measured resistivity.

As mentioned by the referee, ΔV_x in the figures *except* for FigS1a in the original version was defined as the amplitude of the V_x reduction resulting from the randomization of the nuclear spin polarization. On the otherhand, line profiles shown in Fig. S1a was measured at $B = 0$ T without nuclear spin polarization. We plotted Fig. S1 after subtracting the value of V_x obtained outside the Hall bar. Therefore, we revised the title of the vertical axis has been revised into "Subtracted V_x " in the corresponding figure, and renumbered the figure as Fig. S2 in the revised version. The corresponding description has been also added in Supplementary S2. Fig S1a

The experimentally observed NEGATIVE Overhauser field BN as a result of DNP in the breakdown regime (is this a consequence of the negative value of the electronic g-factor in GaAs and the spin flip-flop process?) should be explained in more detail.

To clarify that the negative Overhauser field is caused by the negative value of the electronic g-factor, detailed explanation has been added to "Resistive detection of NR" in the Methods section, as well as in Supplementary S3.

REPLY TO REVIEWER #3:

We appreciate the reviewer's suggestions which have helped further elucidate our findings. We have provided point-by-point replies below in blue.

In the revised version of the paper the authors, in most part, adequately addressed my questions. I think the paper can be published, with some revisions. There are two suggestions I would like to make.

1) The graph given in Fig. R1 could be included in the Supplement. As far as I see, the response is not quite linear, and I cannot really agree with the authors that the linear regime corresponds to the RF power below 1 mW. While this non-linearity does not seem to affect the main conclusions of the paper, I think it is important to make this point clear for the audience: include the corresponding explanation into the main text, and present Fig. R1 so that the interested readers can see the linearity/nonlinearity of the measurement.

According to referee's suggestion, we have added a new section "S9. RF power dependence of the NR intensity" to Supplementary materials with the original Fig. R1. Accordingly, we added an explanation that the NR intensity mapping was performed in the nonlinear regime.

2) I am not completely sure what do the authors mean when they claim that "...the spatial distribution of the DNP strongly links the local QH electron states" (see the authors' Reply and the added sentences in the last paragraph of the main text). I think it would be useful if the authors explained this point in more detail.

In NR intensity mapping, the filling-factor-dependent patterns attributed to DNP were observed in the expected locus of the incompressible strip, which is one of the important characteristics of the quantum Hall effect. To clarify this point, the sentence in the last paragraph of the main text has been rewritten as "The spatial distribution of the DNP is correlated with the locus of an incompressible QH strip."

I think, with the above-mentioned explanations, the manuscript can be published.

We hope that the revised manuscript is now suitable for publication in *Nature Communications*.

REPLY TO REVIEWER #4:

We appreciate the reviewer's positive comments. We have provided point-by-point replies below in blue.

I have read the revised manuscript in light of the previous round of reviews and found that the authors adequately answered the referees' requests. The revised manuscript is much more readable. I think the results are sound and should ultimately be published.

We appreciate the positive comments of the Reviewer and hope that the following revision is sufficient for publication in *Nature Communications*.

However, the paper appears to have been written for PRL; for a general audience such as that of Nature Communications, which will not be familiar with resistively detected NMR, some additional explanations are needed. First of all, why write NR, and not NMR? After all, it is a *magnetic* resonant effect, no? (Or did I miss something?)

In our technique, nuclear resonance is induced through electric quadrupole moments, instead of magnetic dipole moments used for NMR. This technique has been conventionally called "electrically induced nuclear resonance" or "nuclear electric resonance (NER)" [1-4]. To avoid confusion with conventional NMR, we use nuclear resonance (NR), which covers a wide range of techniques including NMR and NER.

[1] E. Brun *et al.*, Phys. Rev. Lett. **8**, 365 (1962).

[2] T. Kushida, and A. H. Silver., Phys. Rev. **130**, 1692 (1963).

[3] D.C. Newitt and E.L. Hahn, Bull. Magn. Resonance **16**, 127 (1944).

[4] M. Ono *et al.*, Appl. Phys. Exp. **6**, 033002 (2013).

In any case, the main change that is needed is to include a supplementary section describing the physics of the technique used. I suggest the authors include a section static the relevant Hamiltonian being probed, and how readout and manipulations are done, and how this relates to current. For RD-NMR people, this is obvious, but for NCOMM readers, this is not obvious at all.

According to the Referee's suggestions, we added new two sections (S1 and S3) to the Supplementary materials. In the first section (S1), starting from the electric quadrupole Hamiltonian, we prove that the RF electric field induces electric quadruple transition

used for nuclear resonance (manipulation). In the second section (S3), we explain that the hyperfine Hamiltonian leads to current-induced dynamic nuclear polarization as well as the resistive detection (read out) of both nuclear and electron spin polarization.

SUMMARY OF CHANGES

All changes have been marked in red to help Reviewers locate them.

- The sentence in the last paragraph (conclusion part) of the main text has been rewritten to answer the Reviewer's inquiry.

- Supplementary Materials discussing electric quadruple transition (S1), dynamic nuclear polarization and its RD detection (S3), and RF power dependence of the NR intensity (S9), have been added according to the Reviewers' remarks. The corresponding brief descriptions have been also added to "Local NR excitation" and "Resistive detection of NR" sections in Methods and in "NR-intensity mapping" section of the main text. All the sentences in Supplementary materials have been correspondingly renumbered.

- The label of the vertical axis of the line profile (Fig. S2a) has been revised and corresponding description has been added to the text (Supplementary S2) to answer the Reviewer's query.

-Minor additional changes in the text have been made to eliminate grammatical errors.

Reviewers' Comments:

Reviewer #1 (Remarks to the Author):

The author addressed my questions of the last round in a satisfying way.

Reviewer #3 (Remarks to the Author):

After the changes requested by the referees are made, the manuscript can be published in Nature Communications.

Reviewer #4 (Remarks to the Author):

I commend the authors for their excellent work. I recommend the paper for publication and look forward to more studies by this research group.